# Research on Technological Innovation Capability of Yancheng Prefabricated Construction Industry Based on Patent Information Analysis

Renyan Lu [1], Feiting Shi [2] and Houchao Sun [2],*

1 Library, Yancheng Institute of Technology, Yancheng 224051, China; lurenyan@ycit.edu.cn
2 School of Civil Engineering, Yancheng Institute of Technology, Yancheng 224051, China; shifeiting@ycit.cn
* Correspondence: sunhouchao@ycit.edu.cn

**Abstract:** In order to improve the innovation capabilities of Yancheng's prefabricated construction industry, the Dawei Innojoy patent database was used to search the prefabricated construction technology patent literature data of Yancheng and other major cities in the Yangtze River Delta region from 2012 to 2022, and the prefabricated construction patents in Yancheng were analyzed from the number of patent applications. Analysis and research were conducted on trends, application type composition, applicants, technical fields, patent legal status, etc. At the same time, the prefabricated building technology innovation capability evaluation system was constructed, and the factor analysis method was used to compare and analyze the prefabricated building technology patent indicators of Yancheng and major cities in the Yangtze River Delta region. The results show that Yancheng has a small number of patent applications, a small proportion of invention patents, a low patent authorization rate, and a low patent conversion rate, and the industry-university-research chain needs to be opened up. Among the cities in the Yangtze River Delta, Yancheng's comprehensive innovation ability in prefabricated building technology is medium to low, where it lags behind in terms of the scale and quality of technological innovation and ranks at the forefront of technological innovation operations. Based on this, the article puts forward countermeasures and suggestions for Yancheng's prefabricated building technology patent applications from three levels, macro, meso, and micro, in order to achieve efficient innovation and promote the high-quality development of Yancheng's new building industrialization.

**Keywords:** prefabricated buildings; innovation capabilities; patents; factor analysis; Yancheng City; Yangtze River Delta

## 1. Introduction

The report of the 20th National Congress of the Communist Party of China clearly stated that it is necessary to accelerate the construction of a new development pattern and strive to promote high-quality development. The report emphasizes that "promoting green and low-carbon economic and social development is a key link in achieving high-quality development". Prefabricated building is a green transformation method used in the construction industry. It is a method of building that produces prefabricated components in a factory and assembles the components on the construction site. It can not only ensure construction quality but also improve labor efficiency, shorten the construction period, reduce pollution, save energy, and have good development prospects [1,2].

Since the 13th Five-Year Plan, Chinese government departments at all levels have successively issued a series of policies to promote the development of prefabricated buildings and encourage local construction companies to transform in the direction of prefabricated buildings. At present, the construction industry in Yancheng City is in a critical period of transformation, upgrading, and accelerated development. Vigorously promoting prefabricated buildings has become an important starting point for the Yancheng City government

to promote the development of the construction industry. In 2021, the Yancheng City government proposed in the "Yancheng City '14th Five-Year Plan' High-Quality Development Plan for Green Buildings" to accelerate the industrialization of new buildings and vigorously develop prefabricated buildings. However, insufficient technological innovation capabilities are an important factor restricting the vigorous development of prefabricated buildings in Yancheng City [3].

The key to the development of prefabricated buildings is to improve quality and efficiency through technological innovation. As one of the main manifestations of scientific and technological products, patents are considered in academia to be an important symbol for evaluating the level of technological innovation and progress. With the increased emphasis on intellectual property rights, research related to patent information has been expanded and improved. Combined with the development of modern Internet big data information, people can conduct a multi-angle and comprehensive measurement and data analysis of the output of patent documents. In the process of technology development, based on the research and analysis of prefabricated building technology patent information, we applied for prefabricated building technology patents, obtained authorization, and successfully realized technology transformation and applications, achieved efficient innovation, assisted the green and low-carbon development of the construction industry, and promoted new buildings in Yancheng City [4,5].

Prefabricated buildings are the direction of high-quality development in the construction industry. The number of applications and authorizations for prefabricated building technology patents is an important indicator of innovation capabilities. Realizing innovation in prefabricated building technology is conducive to the green and low-carbon development of the industry or region.

## 2. Literature Review

### 2.1. Current Status of Research on Technological Innovation of Prefabricated Buildings at Home and Abroad

In countries with advanced industrialization, the construction system, component standardization, technology integration, and large-scale production of prefabricated buildings are relatively mature. A large number of prefabricated buildings not only solve the housing problem for ordinary residential buildings but are also widely used in large buildings, hotels, etc., accelerating the transformation and upgrading of the construction industry. The development of new materials and construction techniques is sufficient to support the construction of more complex building structures, while also bringing overall economic and environmental benefits. Prefabricated buildings in the United States have basically formed a mature standard system. The *PCI Design Handbook* compiled by the Precast and Prestressed Concrete Institute (PCI) covers the design methods, construction technology, and construction quality control of prefabricated buildings, as well as other industry standards and specifications. It has a very important influence internationally [6]. Sweden has adopted the prefabricated technology system of large-scale concrete precast panels and has gradually incorporated the indicators required for the standardization of prefabricated building components into its industrial standard system [7]. In order to promote the standardization of components, connections, etc., the "Swedish Industrial Standard" (SIS) was formed based on module coordination. Japan has formed a complete prefabricated construction technology and standard system. After the introduction of the "PC" construction method from the United States, new materials and new application technologies have been continuously developed, and prefabricated housing has been vigorously developed through prefabricated components such as exterior wall panels, laminated floor panels, balconies, and stairs, with the componentization rate generally reaching 60% [8]. Germany focuses on the standardization and modularization of residential components, emphasizes building durability, promotes prefabricated product technology and environmentally friendly and energy-saving green assembly technology, and has formed

a complete assembly and technology system. The proportion of residential prefabricated components has reached 94.5% [9].

The mature development of prefabricated buildings abroad is inseparable from the research and development of various parts and components, structural systems, and the support of information technology. Many foreign experts and scholars have conducted research from many perspectives. Kiruthika and Parthiban analyzed and explained the design of post-tensioned prestressed concrete beamless floors in precast prestressed concrete commercial buildings, and elaborated on the advantages of prefabricated structures over cast-in-situ concrete structures [10]. El-Sheikh et al. used the structural finite element analysis software OpenSees (version 1.0) to conduct numerical simulation research on a partially bonded prestressed concrete prefabricated frame structure and analyzed its seismic performance [11]. Ozturan et al. tested the connection performance of three types of precast concrete—cast-in-place, composite welding, and bolted connections—under the same loading mode and test configuration. By comparing the performance parameters, such as energy loss, it was shown that the improved bolted connection could be suitable for high seismic areas [12]. Thomas Olofsson's research shows that the use of VR (virtual reality) technology can create a virtual reality environment, which can better optimize design plans and guide on-site construction [13]. Li and Becerik-Gerber proposed that implanting radio frequency identification (RFID) chips into precast concrete (PC) components can help construction personnel better identify them, collect construction information, and monitor project progress [14].

However, the development level of China's prefabricated buildings lags behind that of other countries. Domestic research and development of prefabricated building technology started late, and the standard technology system is not yet mature. At present, core technologies from developed countries with mature prefabricated construction development are mostly introduced from abroad, and technological innovation is carried out based on the actual development situation of China's construction industry. Chinese experts and scholars have also been conducting relevant research in recent years. Qi and Li proposed the full life cycle of prefabricated buildings on the basis of the traditional building life cycle, including four stages: planning and design, production and manufacturing, construction and operation, and maintenance [15]. Zeng Qiang et al. proposed the "five-in-one" development idea of prefabricated buildings, namely design standardization, production industrialization, construction assembly, decoration integration, and management informatization [16]. In terms of component production, Sun Hong et al. proposed that large-scale intelligent PC component automated production lines are the direction of technological innovation in factory-based production in China [17]. In terms of construction technology, Du and Wang et al. conducted a detailed study on the node connection methods of prefabricated buildings and improved the seismic performance of prefabricated buildings [18]. Information technology such as BIM is also a focus of domestic scholars' attention. Zang used the BIM three-dimensional model as an information carrier to build a collaborative working platform for all project participants. All majors can work in the same model, which greatly improves efficiency [19].

### 2.2. Current Status of Patent Measurement Research at Home and Abroad

Research on patent measurement at home and abroad can be summarized into two aspects: patent indicator research and application research in specific technical fields. In terms of indicators, the patent measurement indicators used by foreign scholars mainly include the number of patents, average number of citations, impact factors, technical intensity, scientific intensity, and other indicators used to measure the patent level [20,21]. The academic community generally believes that the Dutch scholar Kunz was the first to use the time distribution of patent information quantity to study the value of patents [22]. With the development of patent measurement analysis methods, research on the application of patent measurement in various specific fields has gradually become richer, mainly reflected in two scenarios. The first is used to study the technological innovation or economic

development of countries and regions. Narin conducted research and analysis on the GDP output value of the United Kingdom, Japan, Germany, and other countries and the number of authorized patents obtained in the United States, and found that there is a strong positive correlation between GDP output value and the number of patents [23]. The second scenario is applied to the study of a certain industry or specific technical field. For example, Ramani and Looze used patent statistical methods to study and analyze the patents of the United Kingdom, Germany, and France in the fields of biotechnology, such as genetic engineering, chemicals, and biocatalysis, and found that Germany is ahead of the United Kingdom and France in many fields [24]. Gemba and Tamada et al. applied patent measurement to corporate research and studied the patent activity tendencies of large Japanese electrical appliance manufacturers and the degree of corporate dependence on science [25].

China's patent measurement research started relatively late. Luo Shisheng's "Quantitative Analysis and Prediction of Patent Documents" in 1994 was the first domestic study on patent measurement. In terms of research on patent measurement indicators, Qiu gave a high-level overview of the theory of patent measurement. His point of view is to "apply mathematical and statistical methods to patent research to explore and tap its distribution structure, quantitative relationships, change patterns and other intrinsic values" [26,27]; Zhang and Liu et al. focused on patents combined with the theory of innovation, explored templates for innovation difficulties, focused on their institutional sources, and measured and evaluated the effectiveness of innovation capabilities with relevant indicators [28]. Some domestic scholars have used R language and Vosviewer and other methods to analyze the development trends and lifespan of patent applications over the years, as well as the geographical distribution of patented technologies. Based on the application trend, life cycle, and geographical distribution of technology, a visualization method is used to obtain a patent map reflecting this field [29]. In terms of technological innovation in specific fields, Zhang et al. used a spatial econometric model to study the relationship between China's patent innovation and regional economic growth based on the number of patent authorizations in 31 provinces and data reflecting regional economic growth indicators [30]. Luan et al. used the Derwent patent database to conduct a patent quantitative analysis in the fields of nanotechnology, biotechnology, and the electric vehicle industry, and researched the technical hot spots in each field [31–33]. Based on patent analysis, Zhang Guangping summarized the patents for prefabricated buildings, then focused on the patents related to structural systems, and summarized the current situation and future of patents related to prefabricated buildings in my country [34].

There are many statistical techniques used in patent trend analysis, including statistical analysis, time series analysis, technical index analysis, patent classification number analysis, technical theme analysis, and patent citation analysis, etc. These methods have their own characteristics and can be used in combination to complement each other to obtain more comprehensive and in-depth patent trend analysis results.

Through the literature review, we find that there is a lack of comparative analysis in research on the technological innovation of prefabricated buildings. There are studies on technological innovation in prefabricated buildings in the literature, but these studies focus on the methods of technological innovation and the research and development of specific technologies. They do not summarize and analyze the results of technological innovation in different regions or further analyze regional technological innovation capabilities. In addition, the innovation capability evaluation based on patent data is mainly concentrated in the industrial field. A large number of documents only focus on the research and evaluation of the traditional construction industry or corporate technological innovation, while focusing on the prefabricated construction industry in a targeted and specialized manner. There is a relative lack of research on technological innovation research and systematic evaluation.

## 3. Methodology

### 3.1. General Idea of Research

China's Yangtze River Delta region is the highland of China's economic development and technological innovation. Since patents are an important symbol of scientific and technological innovation capabilities, in order to study the innovation capabilities of prefabricated building technology in the Yancheng area, we first searched the patent database to study the status of patent applications for prefabricated building technology in the Yancheng area, and then used factor analysis to compare Yancheng and the Yangtze River Delta region. A comprehensive comparative analysis of the main indicators of prefabricated building patent applications in major cities was conducted, to find gaps in the technological innovation capabilities of prefabricated buildings and put forward improved countermeasures and suggestions.

### 3.2. Evaluation Method Selection and Evaluation Index Construction

#### 3.2.1. Evaluation Method Selection

The factor analysis method is an important method in management research and has been widely used in management, economics, medicine, statistics, and other fields. The use of factor analysis to evaluate the level of technological innovation not only conforms to the scientific assessment of the level of technological innovation using methods such as the analytic hierarchy process and fuzzy evaluation methods, but can also reduce the inaccuracies caused by subjective factors such as expert scoring and issuing questionnaires, as well as limitations affecting the analysis and other issues. This method is simple to operate and easy to apply, and its scope of application does not have many restrictions. This study uses factor analysis to reduce multiple variables into a number of common factors with most of the information, and then reduces the dimensions and calculates the scores of the common factors by matching the relevant weights, thus achieving the goal of objective, scientific, quantitative judgments about technological innovation levels.

The steps of the factor analysis method are as follows:

(1) Correlation tests on variables through the KMO model are conducted to analyze whether the initial variable indicators are suitable for factor analysis. Only after matching verification can the next step of the process begin.

(2) Uncategorizing the variables. Since the variables may not be at the same latitude, this operation is performed in order to reduce the differences in the magnitude of the variables.

(3) The data processed in the previous step are then calculated through the correlation matrix to find the eigenvalues and eigenvectors of the data matrix, so that the correlation between different variables can be analyzed and classified.

(4) Calculating the variance contribution rate of each type of data. The cumulative variance contribution rate should be higher than 60% to scientifically explain and replace all initial variables.

#### 3.2.2. Evaluation Index Construction

Regarding regional technological innovation capabilities, domestic and foreign scholars have proposed many evaluation indicators, and prefabricated buildings are a construction method that has developed rapidly in recent years and has certain particularities. At present, there is not much literature on the evaluation and research of technological innovation capabilities in regional prefabricated buildings. The index selection in this article is based on the literature [35] and combined with the actual situation of prefabricated buildings in the Yangtze River Delta in China. The indicators in the literature are screened and replaced, and work is added to form the final evaluation index system, as shown in Table 1.

**Table 1.** Patent indicators and calculation methods for evaluation.

| First-Level Indicator | Secondary Indicators | Indicator Calculation | Variable Code |
|---|---|---|---|
| Scale of technological innovation | Number of patent applications | Number of patent applications | X1 |
| | Effective number of patents | Number of valid patents | X2 |
| | Amount of invention authorizations | Number of invention patents authorized | X3 |
| Technology innovation operation | Average number of claims | Number of claims/total number of patents | X4 |
| | Patent implementation rate | (Transfer + License + Number of Pledges)/Total Number of Patents | X5 |
| | Patent efficiency | Number of effective patents/number of patent authorizations | X6 |
| Technical innovation quality | Average number of patent IPCs | Total number of IPC categories/total number of patents | X7 |
| | Average number of patent citations | Number of patent citations/total number of patents | X8 |

## 4. Results and Discussion

The Dawei Innojoy patent database was searched with "TAC = (prefabricated building or prefabricated house or prefabricated structure or prefabricated building or prefabricated structure or prefabricated frame or residential industrialization or 'prefabricated member' or 'prefabricated construction' or 'fabricated structure' or 'assembly building')" and AD = [20120101 to 20221231] and AR = Yancheng and IPC = E04" as the search formula, the patent applicant's address was set to Yancheng or a major city in the Yangtze River Delta region of China, and the search was for 2012~. Relevant patent data in 2022, a statistical analysis of Yancheng prefabricated building patent data, and comparative analysis with the prefabricated building patent data indicators of major cities in the Yangtze River Delta were explored to identify gaps in innovation capabilities and propose solutions to existing problems. The aims were to improve the quality of patents and achieve efficient innovation, thereby promoting the high-quality development of new building industrialization in Yancheng City.

### 4.1. Current Status of Yancheng Prefabricated Building Technology Patent Development

4.1.1. Number of Patent Applications and Application Trends

Patents are the most dynamic technological innovation achievements, and their application volume and trends reflect important indicators of technological innovation capabilities. As shown in Figure 1, overall, the patent application development process of Yancheng's prefabricated construction technology industry can be classified into three stages: (1) Slow development stage (2012~2016), during which the annual total volume is relatively smaller and develops slowly. The average annual number of applications does not exceed two, and the number of patent applications in 2015 was only three. (2) Rapid development stage (2017~2020). During this period, the number of annual patent applications is generally on a growing trend, exceeding 21 for the first time in 2019. (3) Steady development stage (2021~2022). At this stage, the annual number of relevant patent applications has grown steadily, reaching a peak of 23 in 2022. This is also consistent with the actual development of the prefabricated construction industry in Yancheng City to a certain extent. It has gone through a budding and nurturing period of slow development, a period of sustained rapid development, and a mature period of stable development. (It takes a certain amount of time from patent application to publication, and invention patents generally take about 18 months. Therefore, the data for 2020 have declined and are for reference only).

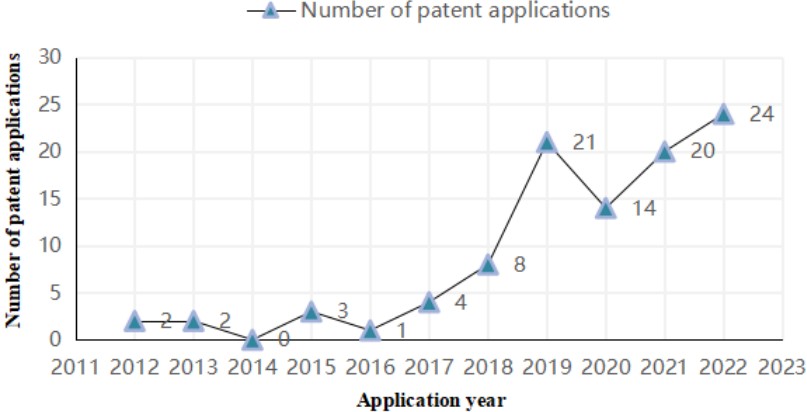

**Figure 1.** Number of patent applications in Yancheng City's prefabricated construction technology industry from 2012 to 2022.

### 4.1.2. Patent Application Type Composition

Patents in my country are divided into three types: invention patents, utility model patents and design patents. Figure 2 shows the composition of patent application types in Yancheng's prefabricated construction technology industry from 2012 to 2022. As can be seen from the figure that utility model patent applications account for the largest number, accounting for 62%, and invention patent applications and authorizations account for 26%. The protection period of utility model patents is only 10 years, but the technical and creative requirements for products are not as high as those for invention patents. After modification, the practical applicability of the products is more prominent, so the number of applications is the largest. Invention patents have a higher degree of legal protection and have the longest protection period, so their value is naturally higher. However, it is more difficult to apply for invention patents. The relative number of applications is much smaller than that for utility model patents, so the number of applications does not account for a high proportion. Comparing the number and proportion of different types of patent applications can represent the R&D strength and innovation capabilities of Yancheng's prefabricated building technology to a certain extent.

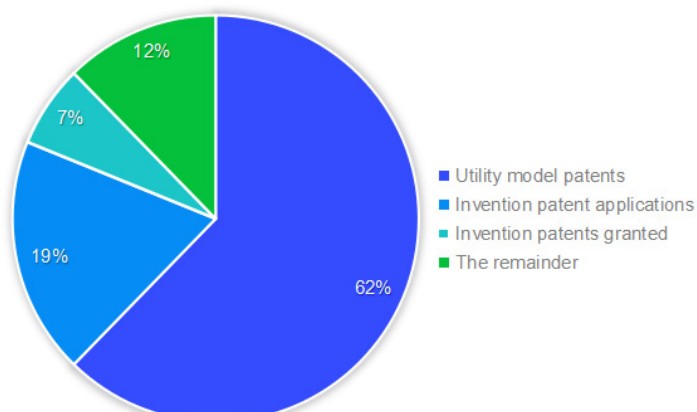

**Figure 2.** Composition of patent application types in Yancheng City's prefabricated construction technology industry.

### 4.1.3. Patent Application Subject

The composition of patent application subjects can analyze the identity composition of innovation subjects. Patent applicants can be divided into basic types such as enterprises, universities, scientific research institutes, and individuals.

Figure 3 shows the main applicants for patents in the prefabricated construction technology industry in Yancheng City. As can be seen from the above figure, the top

10 companies have applied for patents. The top two are the Yancheng Institute of Technology (nine pieces) and Jiangsu Jinmao Technology Development Co., Ltd. (Yancheng, china) companies (seven pieces). The total number of patent applications from these two companies is 16, accounting for about one-third of the total number of patent applications from these 10 units, of which the Yancheng Institute of Technology accounts for nearly 18.37%, and Jiangsu Jinmao Technology Development Co., Ltd. for 14.29%. This shows that Yancheng prefabricated construction technology patent applications are concentrated in Yancheng universities and key prefabricated construction enterprises. Strengthening the cooperation between enterprises and universities in industry, academia, and research is more conducive to the development of industry technology.

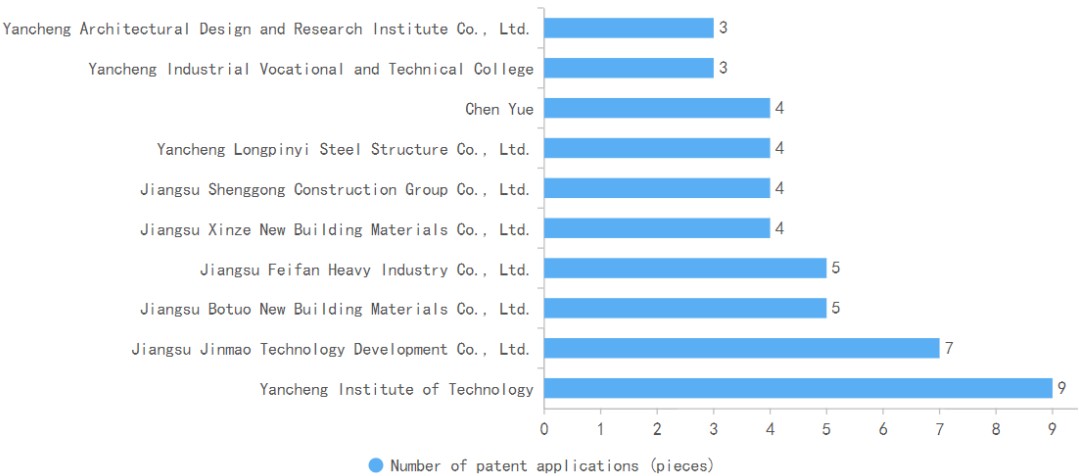

**Figure 3.** The main applicants for Yancheng's prefabricated construction technology industry patents.

4.1.4. Distribution of Patent Technology Fields

There is no specific classification of prefabricated construction technology in the international IPC patent classification table. With the help of the patent database platform and keyword search, the author found a total of 99 patent applications, and 73 were authorized, including 8 invention patents and utility model patents. The author found 65 pieces, among which the top 10 technology categories by number of patent authorizations are shown in Table 2.

It can be found from Table 2 that among the top ten patent categories, the number of patents in the E04B1 direction is far ahead of other technical fields, accounting for 60.27% of the patents, followed by the two fields of E04B2 and E04G21, respectively. The sum of these two fields, which account for 41.1% of the sample, can also be regarded as a technology-intensive area.

In the past five years, despite the impact of the epidemic, the number of patents for prefabricated building technology has grown rapidly. This is inseparable from the Yancheng City government's vigorous promotion of prefabricated building policies. From the perspective of patent technology branches, combined with domestic patent analysis, the research and development of prefabricated building technology focuses on construction technology, building structure systems, component connections, prefabricated components, and other technical directions. Therefore, studying these technology patents, one can accurately grasp the technical hot spots and development trends of Yancheng's prefabricated buildings, enhance the region's technological innovation capabilities in prefabricated buildings, and thereby promote the green and healthy development of Yancheng's construction industry.

**Table 2.** The top ten technical fields in Yancheng's patent authorization volume.

| Serial Number | IPC Classification and Meaning | Number of Patents (Utility Model, Invention Patent)/Pieces | Percentage of Total Sample Patent Data/% |
|---|---|---|---|
| 1 | E04B1 General structure, not limited to walls, such as partition walls, or any structure in floor slabs, ceilings, or roofs | 38, 6 | 60.27 |
| 2 | E04B2 Building walls, such as partition walls and insulating walls, specifically for wall connections. | 18, 1 | 26.03 |
| 3 | E04G21 Preparation, handling, or processing of building materials or building components on site; other methods and equipment used in construction | 8, 3 | 15.07 |
| 4 | E04B5 Floor slab, used for insulated floor slab structures, and its special connectors | 7, 0 | 9.59 |
| 5 | E04C2 Thinner components for building house components, such as various thin plates, flat plates, or panels | 7, 0 | 9.59 |
| 6 | E04C5 Reinforcements, e.g., auxiliary members for concrete | 7, 0 | 9.59 |
| 7 | E04H9 A building or building complex or shelter that can withstand or protect against external anomalies, such as war, earthquakes, or abnormal weather effects | 5, 2 | 9.59 |
| 8 | E04C3 Long-strip structural members for load-bearing | 6, 0 | 8.21 |
| 9 | E04F13 Covering or lining, for example, for walls or ceilings | 4, 0 | 5.48 |
| 10 | E04H5 Industrial and agricultural buildings or building complexes | 3, 0 | 4.11 |

### 4.1.5. Patent Legal Status Analysis

According to the relevant provisions of my country's Patent Law, only authorized patents are protected by law. According to Table 3 and Figure 4, we can see the following: (1) Yancheng City's prefabricated building technology industry applied for a total of 99 patents from 2012 to 2022, of which authorized patents accounted for 73%. (2) The total number of rejections reached nine, accounting for nearly 10%; four were in the substantive examination stage, accounting for 4%, and the number of authorized invention patents accounted for only 8.08%, which fully reflects that invention patents are one of three types of patents. Among the patent types, the application requirements are the highest, the review process is the most stringent, and the review time is the longest. (3) Currently, 65 utility model patents have been applied for and authorized, and 11 more have not paid the annual fee. The relative authorization rate of utility model patent applications in the field of construction technology is relatively high.

**Table 3.** Yancheng prefabricated construction technology industry patent application types and legal status.

| Patent Type | Substantive Examination | Authorization | Withdrawal | Rejection | Disclosure | Annual Fee Not Paid | Total |
|---|---|---|---|---|---|---|---|
| invention patent | 4 | 8 | 4 | 5 | 0 | 2 | 23 |
| utility model patent | 0 | 65 | 0 | 0 | 0 | 11 | 76 |

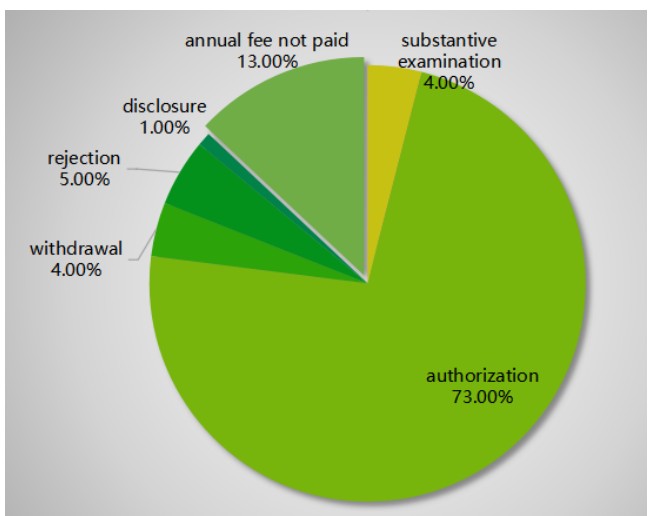

**Figure 4.** Legal status of patent applications for Yancheng prefabricated construction technology industry.

### 4.1.6. Analysis of Patent Development under Regional Collaboration

The integration of the Yangtze River Delta is the national strategy most closely related to Yancheng City, and it has a huge impact on the development of Yancheng's economy and industry. Connecting with the integrated development of the Yangtze River Delta region is a platform for the high-quality development of Yancheng's prefabricated construction technology industry, and it is also an important window period for Yancheng's prefabricated construction technology industry to "accelerate". Through the relevant patent search of the prefabricated construction technology industry in major cities in the Yangtze River Delta, the key cities for patent application and authorization in the prefabricated construction technology industry in the Yangtze River Delta region were obtained. The specific comparison is shown in Figures 5 and 6.

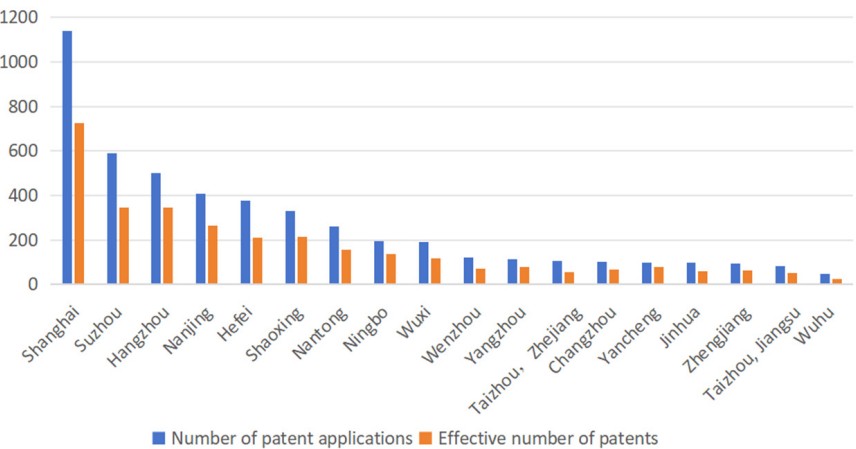

**Figure 5.** Comparison of the total number and effective number of patent applications in the prefabricated construction technology industry in major cities in the Yangtze River Delta.

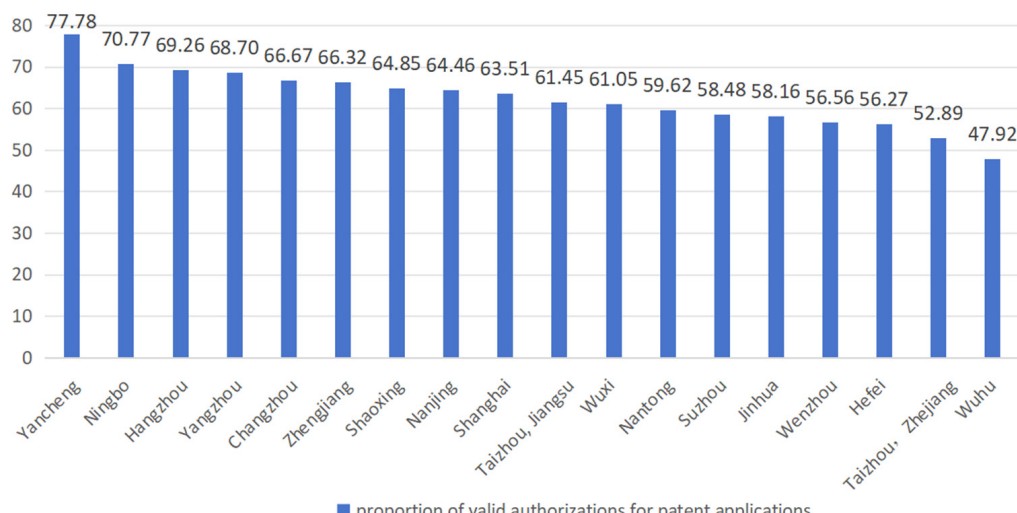

**Figure 6.** Comparison of the proportion of valid authorizations for patent applications in the prefabricated construction technology industry in major cities in the Yangtze River Delta.

According to Figures 5 and 6, the comparative situation of the total number and effective number of patent applications for the prefabricated construction technology industry in major cities in the Yangtze River Delta can be seen: (1) From the perspective of the total number of patent applications, the patents for the prefabricated construction technology industry in the Yangtze River Delta Applications are mainly concentrated in Shanghai, Southern Jiangsu, and Hangzhou and Ningbo. Because Shanghai benefits from its own abundant talent advantages and scientific research institutes, as well as its obviously advantageous financial investment environment, and at the same time relies on the mature and complete industrial chain foundation of the Yangtze River Delta, the total number of applications far exceeds that of other cities, ranking firmly in the list. First, Southern Jiangsu and the Hangzhou–Ningbo regions followed closely behind, relying on their geographical advantages and mature manufacturing base. (2) From the perspective of the total number of patent applications, there is still a certain gap in the development of the prefabricated construction technology industry in the major cities in the Yangtze River Delta, which can be roughly divided into three echelons: First, there are the patents in the four cities of Shanghai, Suzhou, Hangzhou, and Nanjing. The total number of patent applications in the four cities of Shaoxing, Hefei, Nantong, Ningbo, and Wuxi is more than 150 and they belong to the second tier. The total number of patent applications in ten cities including Jinhua, Zhenjiang, Taizhou, and Wuhu is less than 150, belonging to the third echelon. The total number of applications for Yancheng's prefabricated construction technology industry is 99, which is still far behind the second echelon. (3) From the perspective of the proportion of the number of valid patents to the total number of applications, the proportion of valid patents in the field of prefabricated construction technology industry in major cities in the Yangtze River Delta does not completely match the comparison trend between cities in the total number of patent applications. Figure 6 shows that the cities with effective proportions exceeding 60% include 11 cities: Yancheng, Ningbo, Hangzhou, Yangzhou, Changzhou, Zhenjiang, Shaoxing, Nanjing, Shanghai, Taizhou, Jiangsu, and Wuxi. Yancheng ranks first. It can be seen that, although Yancheng City does not have a clear advantage in the total number of patent applications, Yancheng City has a comparative advantage in the proportion of valid patents in the field of the prefabricated construction technology industry, which reflects Yancheng's leading position in prefabricated building technology in the Yangtze River Delta region. The quality potential for innovation in the field is huge.

*4.2. Analysis of Technological Innovation Capabilities of Prefabricated Buildings in Yancheng and Other Major Cities in the Yangtze River Delta Based on Patent Information Analysis*

This study conducts a comparative analysis of the technological innovation capabilities of the prefabricated construction industry in Yancheng and other major cities in the Yangtze River Delta region in China from the perspective of patents. The corresponding patent indicator data of the cities in the sample areas were obtained from patent reports such as Dawei Innojoy patent database, and the data were analyzed using factor analysis processing methods. In SPSS statistics 25, first, each variable indicator was dimensionless and variable significance was carried out in turn. Level analysis, etc., is used to clarify its reasonable weight, and finally the innovation factor score is obtained through expression operation.

4.2.1. Determination of Patent Indicators for Major Cities in the Yangtze River Delta of China

The initial variable data required for patent indicators come from the Dawei patent database. The results of various data in the Yangtze River Delta region after information extraction, integration, and calculation are shown in Table 4.

**Table 4.** Evaluation indicators of technological innovation capabilities of prefabricated buildings in the Yangtze River Delta region.

| City | Number of Patent Applications | Effective Number of Patents | Amount of Invention Authorization | Average Number of Patent IPCs | Average Number of Claims | Patent Implementation Rate | Average Number of Patent Citations | Patent Efficiency |
|---|---|---|---|---|---|---|---|---|
| Yancheng | 99 | 77 | 8 | 1.082 | 6.4 | 0.091 | 0.568 | 0.895 |
| Shanghai | 1140 | 724 | 121 | 1.115 | 6.785 | 0.083 | 0.721 | 0.887 |
| Suzhou | 590 | 345 | 29 | 1.154 | 7.465 | 0.052 | 0.22 | 0.91 |
| Hangzhou | 501 | 347 | 52 | 1.121 | 6.779 | 0.123 | 0.564 | 0.905 |
| Nanjing | 408 | 263 | 74 | 1.175 | 7.302 | 0.131 | 0.842 | 0.867 |
| Shaoxing | 330 | 214 | 25 | 1.075 | 8.208 | 0.153 | 0.818 | 0.871 |
| Hefei | 375 | 211 | 26 | 1.066 | 5.75 | 0.119 | 0.545 | 0.795 |
| Nantong | 260 | 155 | 35 | 1.091 | 8.543 | 0.195 | 0.73 | 0.771 |
| Ningbo | 195 | 138 | 20 | 1.086 | 6.419 | 0.072 | 0.795 | 0.891 |
| Wuxi | 190 | 116 | 25 | 1.103 | 6.156 | 0.096 | 0.56 | 0.833 |
| Changzhou | 102 | 68 | 9 | 1.088 | 5.968 | 0.106 | 0.471 | 0.898 |
| Yangzhou | 115 | 79 | 15 | 1.152 | 5.978 | 0.076 | 0.504 | 0.929 |
| Zhenjiang | 95 | 63 | 9 | 1.064 | 4.938 | 0.065 | 1.191 | 0.861 |
| Wenzhou | 122 | 69 | 25 | 1.043 | 6.207 | 0.246 | 0.783 | 0.945 |
| Taizhou Zhejiang | 104 | 55 | 16 | 1.055 | 6.224 | 0.145 | 0.618 | 0.809 |
| Jinghua | 98 | 57 | 13 | 1.211 | 5.813 | 0.439 | 0.386 | 0.731 |
| Taizhou jiangsu | 83 | 51 | 7 | 1.176 | 5.714 | 0.1 | 0.32 | 0.806 |
| Wuhu | 48 | 23 | 8 | 1.087 | 5.783 | 0.273 | 0.348 | 0.88 |

4.2.2. Data Processing of Patent Indicators in Major Cities in the Yangtze River Delta

In order to use the patent index data of major cities in the Yangtze River Delta region of China to evaluate the innovation capabilities of the prefabricated construction technology industry in Table 4, SPSS 25.0 software was used to conduct a factor analysis, test the reliability of the constraint system, and obtain the results reflecting the assembly performance of each city. Various scores and comprehensive scores and rankings of public factors

of architectural technology innovation were considered. After completing the relevant settings in the SPSS 25.0 software, one has only to run it, and the software will automatically generate the relevant result data.

Suitability Test

KMO and Bartlett tests can measure whether a factor analysis is suitable. From Table 5 below, we can see that the value of the KMO statistic is 0.611 (greater than 0.5), and the SPSS test result shows that the *p*-value is 0 (less than 0.05), which reflects that the selected patent indicator variables can be used to study the relationship between variables using factor analysis. Combining the two experiments, the final result is more convincing.

**Table 5.** KMO and Bartlett test.

| KMO and Bartlett Test | | |
|---|---|---|
| KMO value | | 0.611 |
| Bartlett's test of sphericity | approximate chi-square | 115.738 |
| | degrees of freedom | 28 |
| | significance | 0.000 |

Variance Contribution Rate Analysis

According to the total variance explanation value in Table 6, it can be concluded that some and all of the first three causes are greater than 1, and they reach 3.306, 1.795, and 1.070, respectively, accounting for 41.322%, 22.441%, and 13.371% of the overall variance, respectively. The eigenvalues of the first three factors account for a total of 77.135% of the variance, so the system's selection of the first three common factors is in line with the requirements of the scientific characteristics of the factor analysis method. If you want to simplify the complexity of the original data, you can select these three indicators to compare. Appropriately, they replace almost all the basic information in the original indicator.

**Table 6.** Total variance explained value.

| Element | Initial Eigenvalue | | |
|---|---|---|---|
| | Eigenvalues | Percentage of Variance | Cumulative Percentage |
| 1 | 3.306 | 41.322 | 41.322 |
| 2 | 1.795 | 22.441 | 63.764 |
| 3 | 1.070 | 13.371 | 77.135 |
| 4 | 0.747 | 9.334 | 86.469 |
| 5 | 0.549 | 6.856 | 93.325 |
| 6 | 0.462 | 5.779 | 99.104 |
| 7 | 0.069 | 0.867 | 99.972 |
| 8 | 0.002 | 0.028 | 100.000 |

Common Factor Naming

It can be seen from Table 7 that each starting factor can appropriately represent the original indicator, so there is no need to increase the process of factor rotation. Public factor 1 has relatively large loadings on the number of patent applications, the number of valid patents, the number of invention authorizations, and the average number of claims. Because the four indicators represent the scale and overall influence of patents, public factor 1 is represented by F1, that is, the technological innovation scale factor; public factor 2 has a relatively large load on the patent implementation rate and patent efficiency, because the two indicators indicate the patent operation situation, so public factor 2 is

represented by F2, which is the technological innovation operation factor; public factor 3 has a relatively large load on the average number of patent IPCs and the average number of patent citations, because both indicators use patent coverage as the basis to show the strength of patent quality, so the author uses F3 to represent common factor 3, that is, the technology innovation quality factor.

**Table 7.** Factor loading coefficient matrix values.

| Index | Factor Loading Coefficient | | |
|---|---|---|---|
| | **Factor1** | **Factor2** | **Factor3** |
| Number of patent applications | 0.940 | 0.233 | 0.101 |
| Effective number of patents | 0.943 | 0.251 | 0.089 |
| Amount of invention authorizations | 0.943 | 0.103 | −0.019 |
| Average number of claims | 0.607 | −0.178 | −0.132 |
| Patent implementation rate | −0.141 | −0.804 | 0.151 |
| Patent efficiency | 0.056 | 0.838 | −0.089 |
| Average number of patent IPCs | 0.208 | −0.263 | 0.794 |
| Average number of patent citations | 0.173 | 0.023 | −0.876 |

Factor Scores and Rankings

Table 8 shows the formulas of the three common factors, which can be edited as follows (relational expressions are established based on standardized data).

$$F1 = 0.288X1 + 0.287X2 + 0.312X3 + 0.068X4 + 0.239X5 + 0.050X6 + 0.099X7 − 0.087X8 \tag{1}$$

$$F2 = 0.063X1 + 0.073X2 − 0.053X3 − 0.072X4 − 0.233X5 − 0.534X6 − 0.166X7 + 0.581X8 \tag{2}$$

$$F3 = 0.071X1 + 0.065X2 − 0.041X3 + 0.521X4 − 0.158X5 − 0.030X6 − 0.643X7 + 0.085X8 \tag{3}$$

**Table 8.** Component score coefficient matrix.

| Index | Element | | |
|---|---|---|---|
| | **Element 1** | **Element 2** | **Element 3** |
| Number of patent applications | 0.288 | 0.063 | 0.071 |
| Effective number of patents | 0.287 | 0.073 | 0.065 |
| Amount of invention authorizations | 0.312 | −0.053 | −0.041 |
| Average number of patent IPCs | 0.068 | −0.072 | 0.521 |
| Average number of claims | 0.239 | −0.233 | −0.158 |
| Patent implementation rate | 0.050 | −0.534 | −0.030 |
| Average number of patent citations | 0.099 | −0.166 | −0.643 |
| Patent efficiency | −0.087 | 0.581 | 0.085 |

The public factor score obtained above and its corresponding variance contribution rate are used to obtain the overall comprehensive score through a weighted algorithm: F = (39.086F1 + 19.735F2 + 18.314F3)/77.135 (where F represents the final index score).

The final calculation results are shown in Table 9.

**Table 9.** Comprehensive score table for technological innovation capabilities of prefabricated buildings in major cities in the Yangtze River Delta region.

| City | Factor Score | | | | | | Comprehensive Score | Comprehensive Ranking |
|------|------|------|------|------|------|------|------|------|
| | $F_1$ | | $F_2$ | | $F_3$ | | | |
| | Score | Ranking | Score | Ranking | Score | Ranking | | |
| Shanghai | 2.932 | 1 | 0.706 | 4 | 0.094 | 8 | 1.689 | 1 |
| Suzhou | 0.677 | 5 | 1.117 | 2 | 1.669 | 1 | 1.025 | 2 |
| Hangzhou | 0.800 | 3 | 0.590 | 7 | 0.394 | 6 | 0.650 | 3 |
| Nanjing | 1.191 | 2 | −0.352 | 14 | −0.023 | 10 | 0.508 | 4 |
| Yangzhou | −0.716 | 13 | 1.158 | 1 | 0.934 | 4 | 0.155 | 5 |
| Taizhou, Jiangsu | −0.797 | 15 | −0.070 | 11 | 1.550 | 2 | −0.054 | 6 |
| Shaoxing | 0.583 | 6 | −0.443 | 15 | −1.172 | 17 | −0.096 | 7 |
| Hefei | −0.034 | 7 | −0.189 | 12 | −0.186 | 11 | −0.110 | 8 |
| Wuxi | −0.308 | 9 | 0.074 | 10 | 0.086 | 9 | −0.117 | 9 |
| Ningbo | −0.277 | 8 | 0.609 | 5 | −0.683 | 13 | −0.147 | 10 |
| Changzhou | −0.858 | 17 | 0.803 | 3 | 0.276 | 7 | −0.164 | 11 |
| Yancheng | −0.727 | 14 | 0.595 | 6 | −0.309 | 12 | −0.290 | 12 |
| Nantong | 0.764 | 4 | −1.800 | 17 | −1.031 | 15 | −0.318 | 13 |
| Wuhu | −0.989 | 18 | −0.200 | 13 | 0.528 | 5 | −0.427 | 14 |
| Wenzhou | −0.529 | 11 | 0.266 | 9 | −1.106 | 16 | −0.463 | 15 |
| Jinhua | −0.311 | 10 | −2.842 | 18 | 1.512 | 3 | −0.526 | 16 |
| Taizhou, Zhejiang | −0.564 | 12 | −0.457 | 16 | −0.692 | 14 | −0.567 | 17 |
| Zhenjiang | −0.837 | 16 | 0.435 | 8 | −1.842 | 18 | −0.750 | 18 |

Comprehensive Comparative Analysis of Innovation Capabilities between Yancheng and Other Major Cities in the Yangtze River Delta

From Table 8, we can see that there is a positive correlation between a city's comprehensive score, comprehensive ranking, and technological innovation capabilities. There are positive and negative signs for the three public factors and the comprehensive score. A positive value means that the technological innovation capability in this aspect is higher than the average level, and a negative value means that the technological innovation capability in this aspect is below the average level. From the observation of the comprehensive scores, we can see that there are differences between cities. The comprehensive scores of five cities, including Shanghai, Suzhou, Hangzhou, Nanjing, and Yangzhou, are above the average level, while the comprehensive scores of the remaining cities, including Yancheng, are below the average level. The range of comprehensive scores between most regions is not large, lying in the (−1, +2) interval.

Only two of the five cities with a comprehensive score above the average have a comprehensive score greater than 1, namely Shanghai and Suzhou. Their overall data scores are 1.689 and 1.025, indicating the technological innovation of prefabricated buildings among these five cities. There are also gaps in capabilities. In addition, among these cities, Shanghai ranks first in the technological innovation scale factor, Yangzhou ranks first in the technological innovation operation factor, and Suzhou ranks first in the technological innovation quality factor data score. Yancheng ranks sixth in the technological innovation operation factor data and is at the forefront of the cities in the Yangtze River Delta. The data scores of the technological innovation scale factor and technological innovation quality factor rank 14th and 12th, respectively, which is lower than the average position. This shows that the development of technological innovation capabilities for prefabricated

buildings in cities in the Yangtze River Delta is unbalanced, and Yancheng City is generally in a backward position.

The technological innovation scale factor has the largest variance contribution rate after calculation, which is 41.322%. It is the main aspect that affects the technological innovation capability of the prefabricated construction industry. Only six regions scored above the average on this public factor. Among them, Shanghai, Nanjing, and Hangzhou scored the highest, with scores of 2.932, 1.191, and 0.800, respectively. The above means that these three regions are at the top of the overall data ranking. Yancheng's technological innovation scale score is −0.727, ranking 14th among the 18 cities in the Yangtze River Delta. Therefore, if we want to improve Yancheng's prefabricated building technological innovation capabilities, we should continue to increase the scale of patents and expand patented products.

The variance contribution rate of the technological innovation operation factor is 22.441%. There are 10 regions with scores above the average level on this common factor. Among them, Yangzhou and Suzhou scored high, 1.158 and 1.117, respectively. A higher ranking in the technological innovation operation factor also improves the overall ranking of these two regions. Yancheng's technological innovation operation factor score is 0.595, ranking sixth, which has improved the overall ranking of urban innovation. It can be seen that the operation of patents is important. However, many regions have ignored this point. For example, Nanjing and Shaoxing, which are ranked high in the comprehensive rankings, rank relatively low in terms of technological innovation operation factors. Therefore, regions need to strengthen technological innovation operation management.

The variance contribution rate of the technological innovation quality factor is 13.371%. There are nine cities whose scores on this common factor are above the average level. Among them, Suzhou, Taizhou, and Jinhua have the highest scores, with scores of 1.669, 1.550, and 1.512, respectively. It is obvious that Taizhou's high ranking in the technological innovation quality factor has also boosted the region's overall ranking, even though its scores on the other two factors are lagging behind. Yancheng's technological innovation quality factor score is −0.309, ranking low. Therefore, improving the quality of patents will help improve industrial technological innovation capabilities.

The successful implementation of prefabricated building patents requires not only an increase in the number of patent applications and authorizations, but also an improvement in the quality of patent technology innovation. Prefabricated building patent technology has innovative integrated design, ensures the safety and durability of building structures, simplifies the on-site assembly process, achieves the successful transformation of regional patent technology, forms economies of scale, significantly reduces construction costs, and improves the innovation capacity of regional prefabricated building technology. The protection and renovation project of Blythe House in London, UK, uses prefabricated building technology to solve the challenges of traditional building renovation, while preserving the cultural value of historic buildings. The "Sky City" prefabricated building of the Yuanda Technology Group in Changsha, China, has the advantages of prefabricated buildings in speed, cost-effectiveness, and sustainability. Yancheng should not only maintain the implementation rate and efficiency of prefabricated building patents, but also increase the scale and quality of prefabricated building technology innovation, and increase the transformation of patent technology achievements, so as to comprehensively improve the innovation capacity of prefabricated building technology in Yancheng and realize the green and sustainable development of the construction industry in Yancheng as soon as possible.

## 5. Conclusions and Suggestions

### 5.1. Main Conclusion

By analyzing the patent document data of the prefabricated construction technology industry in Yancheng City and major cities in the Yangtze River Delta from 2012 to 2022, the following conclusions can be drawn:

(1) Judging from the current development status of prefabricated building patents in Yancheng City, the number of patent applications in Yancheng City has been at a low level for a long time, the proportion of invention patents is relatively small, and the patent authorization rate is not high. After 2016, the number of patent applications increased significantly and entered a rapid climbing stage. In the prefabricated building technology branch, patent applications in Yancheng City focus on component, structure, and construction technology patents. Judging from the top 10 IPC classification numbers with patent authorizations, the E04B1 and E04B2 patents are the main ones, which is an advantage for Yancheng City's construction industry. In the field of prefabricated building technology industry, the number of patent applications in Yancheng City is stable. The top two applicants are in an absolutely dominant position. Their patent applications account for one-third of the applications of the top 10 applicants, of which the Yancheng Institute of Technology accounts for 18.37% and Jiangsu Jinmao Technology Development Co., Ltd. accounts for 14.29%. However, the major patent applicants are well-known enterprises or key universities in the city's industry. Small and medium-sized enterprises and high-tech enterprises have fewer patent applications, which limits the development space of small and medium-sized enterprises to a certain extent. The proportion of individual applications is slightly higher, and there is little cooperative research and development. In the long run, it will not be conducive to the integration and healthy development of the industry from the perspective of the legal status of patents. The number of utility model patent authorizations is relatively large, eight times that of invention patent authorizations. It is difficult to authorize invention patents, and there are problems such as a failure to pay annual fees after obtaining invention patent authorization. Invalidation, as well as the small number of invention patent transfers, and the integration of industry, academia, and research have not yet truly taken shape.

(2) From the perspective of major cities in the Yangtze River Delta, the number of patent applications and authorizations in the field of the prefabricated construction technology industry in Yancheng City is significantly lower, far behind the patent development level of Shanghai, Hangzhou, Nanjing, Hefei, Suzhou, Nantong, and other countries. Obviously, Yancheng City has a comparative advantage in the effective proportion of patent authorization applications in the field of prefabricated building technology industry, which to a certain extent reflects Yancheng City's great potential for innovation quality in the field of prefabricated building technology in the Yangtze River Delta.

(3) The authors have analyzed and studied the technological innovation capabilities of prefabricated buildings in Yancheng and major other cities in the Yangtze River Delta based on the perspective of patents. Drawing lessons from domestic and foreign patent index systems, we designed three first-level indicators and eight second-level indicators around patent information planning: technological innovation scale, technological innovation quality, and technological innovation operation. We combined the factor analysis method to build a model for rationality and correlation analysis, furthermore extracting three common factors. Through a comparative analysis of the technological innovation capabilities of prefabricated buildings in Yancheng and major cities in the Yangtze River Delta, the specific situation of Yancheng's technological innovation capabilities in prefabricated buildings among the cities in the Yangtze River Delta based on patent information analysis was obtained. Generally speaking, the technological innovation capabilities of prefabricated buildings are medium to low. Among them, the scale and quality of technological innovation are lagging behind, and technological innovation operations are at the forefront.

## 5.2. Suggestions

(1) In view of the current situation that Yancheng's prefabricated building technology innovation capabilities are not strong, government departments need to improve the

patent management system, such as by optimizing the patent application incentive system, establishing a patent application evaluation system, and strengthening the patent application supervision system. There needs to be a focus on the quality of patent applications and the transformation of the results, such as establishing a high-value patent cultivation demonstration center, conducting patent micro-navigation retrieval and analysis, and building a patent data-sharing service platform. In this way, we can strengthen the construction of prefabricated building technical talent teams and promote the cultivation and introduction of key prefabricated building enterprises.

(2) In the future, we can carry out research on how to effectively transform patents for prefabricated building construction, and the contribution of patent technology innovation to the high-quality development of the regional construction industry in order to enhance regional prefabricated building technology innovation capabilities.

**Author Contributions:** Conceptualization, H.S. and R.L.; methodology, R.L.; software, R.L.; validation, R.L., F.S. and H.S.; investigation, F.S.; resources, F.S.; data curation, R.L.; writing—original draft preparation, R.L.; writing—review and editing, H.S.; visualization, R.L.; supervision, F.S.; project administration, H.S.; funding acquisition, R.L. All authors have read and agreed to the published version of the manuscript.

**Funding:** This research was funded by the Yancheng Natural Science Soft Project Foundation of China, grant number yckxrkt2023-44.

**Institutional Review Board Statement:** Not applicable.

**Informed Consent Statement:** Not applicable.

**Data Availability Statement:** The original contributions presented in the study are included in the article, further inquiries can be directed to the corresponding author.

**Acknowledgments:** The authors gratefully acknowledge the financial support of YCIT.

**Conflicts of Interest:** The authors declare no conflicts of interest.

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
