# Peer review of "Research on Technological Innovation Capability of Yancheng Prefabricated Construction Industry Based on Patent Information Analysis"

_buildings, doi:10.3390/buildings14092968_

Round 1
Reviewer 1 Report
Comments and Suggestions for Authors
In the literature review part it is better to include the implemented statistical techniques on the trend analysis of the patents.
The Heading “Introduction” is not numbered. The following heading numbers should be incremented by 1.
The authors use phrases like “In foreign countries with developed industrialization levels…” The manuscript is written to be published in an international journal. Therefore phrases like “foreign countries” are not suitable descriptions. “Developed countries” etc. can be used instead. Similarly “production in my country”,
In page 2, the abbreviation “PC” should be written in long form for the first time use.
In page 3. Instead of “Li and Becerik-Gerber believe …” it is better to write “Li and Becerik-Gerber puts forward…”
In page 7, there is a typo “26%. %.” Please correct it.
In Page 8 related with Figure 3. The figure caption is written as “Number of patent” this confuses the reader. Do the given numbers represent number of patents or the number of patent applications? Please clarify this by adding explanation to the text. Moreover, the number of engineers and specialists working at the R&D departments at the given companies can be given to measure the efficiency as well.
Page 9 Table 3. There is a typo “invention paten” please correct it.
Page 10. Please correct the typo “figure5 and 6”
Page 11. “According to Figure 6 and Figure 7,” should be corrected as “According to Figure 5 and Figure 6,” Moreover Figure 7 is also used in the text separately please check it.
In page 11 and 12. The utilized software SPSS 22.0 or 25.0 please check it. It is confused.
In page 15 Please correct the typo “lower than average. position." The first “.” Should be removed.
In the discussion part it is better to compare the budgets of the R&D departments of the companies. It may also have important effect on the number of patents.
Comments on the Quality of English LanguageThe authors use phrases like “In foreign countries with developed industrialization levels…” The manuscript is written to be published in an international journal. Therefore phrases like “foreign countries” are not suitable descriptions. “Developed countries” etc. can be used instead. Similarly “production in my country” is better to be revised.
Reviewer 2 Report
Comments and Suggestions for Authors
The comments are placed in the attachment.
